# Differential Co-Expression Analyses Allow the Identification of Critical Signalling Pathways Altered during Tumour Transformation and Progression

**DOI:** 10.3390/ijms21249461

**Published:** 2020-12-12

**Authors:** Aurora Savino, Paolo Provero, Valeria Poli

**Affiliations:** 1Molecular Biotechnology Center, Department of Molecular Biotechnology and Health Sciences, University of Turin, Via Nizza 52, 10126 Turin, Italy; 2Department of Neurosciences “Rita Levi Montalcini”, University of Turin, Corso Massimo D’Ázeglio 52, 10126 Turin, Italy; paolo.provero@unito.it; 3Center for Omics Sciences, Ospedale San Raffaele IRCCS, Via Olgettina 60, 20132 Milan, Italy

**Keywords:** differential co-expression, biological networks, cancer, tumour progression, computational biology, bioinformatics

## Abstract

Biological systems respond to perturbations through the rewiring of molecular interactions, organised in gene regulatory networks (GRNs). Among these, the increasingly high availability of transcriptomic data makes gene co-expression networks the most exploited ones. *Differential* co-expression networks are useful tools to identify changes in response to an external perturbation, such as mutations predisposing to cancer development, and leading to changes in the activity of gene expression regulators or signalling. They can help explain the robustness of cancer cells to perturbations and identify promising candidates for targeted therapy, moreover providing higher specificity with respect to standard co-expression methods. Here, we comprehensively review the literature about the methods developed to assess differential co-expression and their applications to cancer biology. Via the comparison of normal and diseased conditions and of different tumour stages, studies based on these methods led to the definition of pathways involved in gene network reorganisation upon oncogenes’ mutations and tumour progression, often converging on immune system signalling. A relevant implementation still lagging behind is the integration of different data types, which would greatly improve network interpretability. Most importantly, performance and predictivity evaluation of the large variety of mathematical models proposed would urgently require experimental validations and systematic comparisons. We believe that future work on differential gene co-expression networks, complemented with additional omics data and experimentally tested, will considerably improve our insights into the biology of tumours.

## 1. Biological Networks

Biological systems are complex in nature, their behaviour being governed by the interactions of many molecular components (e.g., coding and non-coding RNAs, proteins), through several regulatory layers [1] (e.g., promoter binding, miRNA–mRNA interaction, post-translational modifications). Cancer is no exception, being the result of multiple perturbations within a single cell that also affect cell–cell and cell–microenvironment communication. Each perturbation does not act in isolation but is influenced and in turn influences the whole system, with reciprocal relationships occurring between most components [2].

Therefore, accurately depicting a biological system such as a cancer cell requires knowledge about elements’ interactions, and in particular about the regulatory layer described as gene regulatory networks (GRNs). Gene regulatory networks, like all biological networks, represent biological components as nodes and their interactions, either physical or functional, as edges (Figure 1). GRNs are the ideal reconstruction of interactions between genetic elements, comprising the activity of transcription factors (TFs) on their targets’ expression, post-translational modifications influencing a protein’s impact on other elements of the network, epigenetic modifications altering transcription and many additional levels of regulation. While the overall goal of most biological network studies is the inference of GRNs, this is a complex and laborious task that is generally approached by setting some simplifying assumptions and by analysing one kind of relationship at a time, based on physical or other kinds of interactions, as described below.

One caveat is that, despite the general assumption that physical or genetic interactions indicate shared functions or belonging to the same molecular pathway(s), this is not necessarily true, since the information used for edges’ inference is not a direct measure of a functional relationship. Moreover, each network type estimates only part of the overall GRN structure (e.g., transcriptional regulation, protein–protein interactions), losing the information hidden to the specific data type used for its construction, which could be revealed by combining the results obtained with additional data sources.

The most frequently studied biological networks based on physical interactions are protein–protein interaction (PPI) networks, where nodes are proteins and links indicate direct binding, but the binding of a TF to its targets’ promoters can also be represented in a network. A particular type of biological network is pathways, curated and deposited in repositories such as KEGG (Kyoto Encyclopedia of Genes and Genomes) [3], where nodes can be either proteins or small molecules and edges indicate a variety of interactions, among which are enzymatic reactions. Genetic interactions such as synthetic lethal interactions can also be studied as networks [4], assuming them to indicate that the two genes belong to the same pathway. Additionally, methods have been developed to build a variety of biological networks based on metabolic, single nucleotide polymorphisms (SNPs) and phenotypic data (reviewed in [5]).

High-throughput gene expression assays are often used to infer functional relationships between genes from correlations between their expression levels, building the so-called gene co-expression networks (Figure 1). The potential relevance of this method to estimate GRNs is supported by the knowledge that genes with similar transcriptional expression profiles are likely to be regulated through the same mechanisms and to participate in the same functions, or to physically interact [6,7,8,9]. This information can also be combined with other information such as, for example, transcription factor binding and/or PPIs, to obtain a more complete and accurate representation of molecular elements’ interrelationships [10]. Specific open-access databases have been created to store physical and functional networks. Some of the most popular are summarised in Table 1.

## 2. Gene Co-Expression Networks

Gene networks have proved to be a valuable tool to understand some general principles governing biological systems, revealing that gene co-expression is organised in a non-random fashion, with highly interconnected genes (hubs) and groups of tightly co-regulated genes (modules) (Figure 1). These principles were first identified in simple unicellular organisms such as yeast, but were then demonstrated to be preserved across evolution.

Seminal studies on *Saccharomyces cerevisiae* have established that there is a relationship between genes’ topological features in the network and their function [18,19]. In fact, they have shown that biological networks have a scale-free topology, with many genes having few interactors and a few genes having many interactors (central nodes or hubs) [18] (Figure 1). Centrality distinguishes the most important nodes for a network’s integrity: hub genes’ removal has the strongest effect in inducing lethality [18,19]. This has important implications for the ability to predict the effect of interfering with specific genes on the resulting biological phenotype.

The same concepts have been extended to other organisms, including humans, and applied to the study of disease [20,21,22,23]. In Mendelian disorders, for example, genes harbouring disease mutations with a dominant phenotype display significantly higher network connectivity than genes with a recessive phenotype [23,24], and cancer driver genes tend to be centrally located in protein–protein interaction networks [22]. These observations give important indications for genes’ function prediction and for the prioritisation of therapeutic targets [25,26].

Biological networks are intrinsically modular [27], with genes belonging to the same module usually sharing common functions [28]. Identifying modules across multiple networks can help in discovering patterns, and shed light on the underlying mechanisms of biological systems. In particular, much interest has been directed to the identification of shared modules in different conditions, and also in cancer biology [29].

Therefore, the organisation of gene co-expression networks can inform about genes and biological processes that are relevant for cell behaviour, facilitating predictions for specific gene knock-downs (KDs) and drug treatments’ effects.

However, gene and protein networks provide a static snapshot of molecular interactions within a tissue, while biological systems are highly dynamic. Perturbations caused by evolutionary changes, disease, environmental stresses and oncogene activation result in rewiring the network’s topology [30,31]. A simple example of changes in co-expression is Max, a transcription factor, which acts either as an activator or a suppressor on the same target genes depending on whether it binds to Myc or Mad [32]. Accordingly, Max expression exhibits either positive or negative correlation with its targets depending on the conditions. The study of such changes can give valuable insights on disease mechanisms and altered pathways [33], providing information on those weakly differentially expressed genes that are regulated at a different level [34]. In fact, several reports confirmed that differentially expressed and differentially co-expressed (DC) genes tend not to overlap, and hence reflect distinct regulatory processes [35]. Additionally, the differential expression of genes with low levels of expression, such as TFs, might be difficult to retrieve, while it could be evidenced by the coherent change in co-regulation of a group of its targets, identified as a DC module. These motivations led to the definition of several methods for differential co-expression analysis, aimed at identifying changes in networks’ structure across conditions [36]. Similarly, differential network methods have been successfully applied to other types of molecular network models, including genetic interaction networks [37] and PPI networks [38].

Finally, the integration of various data types can build upon co-expression networks to infer the sources of variability responsible for the observed differential co-expression. As represented in the simplified model of Figure 2, the loss of co-expression between a transcription factor and its target can be accounted for by intervening regulatory mechanisms, such as the transcription factor post-translational modifications, mutations and its association with alternative co-factors or epigenetic control (DNA methylation, non-coding RNAs), not necessarily leading to a detectable differential expression at the mRNA level. Adding this information to the model helps, on one side, to explain the molecular processes involved in the co-expression change and, on the other side, to identify the functional correlates of less experimentally accessible regulatory layers (e.g., protein phosphorylation).

Here, we will provide a brief overview of the main methods used for differential co-expression analysis, and we will then focus on the results achieved by such methods in the study of cancer biology and tumour progression. With respect to previous reviews on differential co-expression [36,39], our work specifically focuses on biological insights that these methods have allowed in cancer biology. Moreover, we extend the discussion of van Dam and co-authors on differentially co-expressed modules [39], by including additional “global” and “gene-specific” levels of network analysis, as detailed below.

## 3. Methods for Differential Co-Expression Analysis

The main methods proposed for the study of differential co-expression can be distinguished into three broad classes, based on the principles they rely on and on their application (Table 2, Table 3 and Table 4, Figure 3):“global network” approaches aim at reconstructing the whole differential co-expression network between two or multiple conditions. Global features of the differential network can be studied, such as edge distribution, modularity or entropy.“module-based” approaches aim at identifying groups of co-regulated genes that are differentially interconnected under specific conditions. Usually, differences in connectivity within a module are analysed, but also methods for the identification of pairs of DC modules (connectivity between modules) have been proposed. Additionally, modules can be either identified unbiasedly from data, or pre-specified based on prior knowledge (here defined as “pathway-based” methods).“single-gene” approaches study the change in co-expression between pairs of genes or between a gene and its neighbours in the network. These approaches are particularly suited to select experimentally testable hypotheses. In principle, all “global network” methods can be used to drive “gene-specific” outputs using node-centred metrics that summarise the relevance of a gene within the differential network. On the other side, single genes thus identified can be used as seeds to build differential modules with neighbouring genes within the network.

Moreover, all methods can be distinguished based on the number of conditions they can simultaneously compare, and on the amount of previous information they rely on and they can account for. For example, gene co-expression networks’ structure and differential modules can be obtained in three ways: (i) making use of transcriptional data only in a completely unbiased way (Table 3, “Unbiased” methods); (ii) analysing transcriptional data of pre-defined gene sets such as Gene Ontology lists (Table 3, “A priori” methods) (iii) combining manually curated protein–protein interaction networks or known pathway structures with expression data (Table 3, “Structured pathway” methods), to identify a differential usage of predetermined nodes and edges. This method, while allowing for the reduction the noise, it is unable to uncover new, potentially relevant, interactions. Finally, additional features such as genetic variation may be integrated to improve networks’ interpretation [92,93] (Table 5).

The “module-based” approach to differential co-expression has advantages and downsides. On one hand, identifying DC modules allows for achieving a higher robustness to noise than the “gene-specific” approach. Moreover, the results are easier to interpret than those obtained via the “global network” approach, resulting in smaller gene lists that can be studied for their enrichment in gene ontology or pathways’ signatures. On the other hand, it hardly allows for extracting general principles on cancer’s network behaviour, while not giving, on its own, information about individual cancer targets. Nevertheless, it can be combined with gene prioritisation approaches to obtain both robust and clinically informative results.

The most fine-grained analysis of differential co-expression aims at identifying pairs of genes that change their co-expression across conditions, losing, gaining or switching the sign of their correlation. Often, single genes with overall large changes in connectivity across the whole network are identified. This approach has the advantage of providing easily testable hypotheses, involving one or two specific genes, at the cost of lower interpretability. Therefore, most studies combine the “gene-specific” approach with the identification of modules as ensembles of DC genes or built from a single gene changing its connectivity, which is used as a seed for a module’s construction.

An intriguing approach consists in using co-expression to identify clusters of samples, e.g., subsets of tumours differing in the co-expression of TF targets, leading to the identification of modulators of differential co-expression [62,98,99,100] (Table 6). Most of the algorithms are implemented in R and freely available, to facilitate their usage, together with network visualisation tools (e.g., Cytoscape and Gephi [101,102]).

Usually, these methods take as input two or multiple gene expression datasets that the user wishes to compare, and give as output either a network of dysregulated connections (“global network” approach), a list of genes belonging to a dysregulated module (“module-based” approach) or a ranking of dysregulated nodes (“single-gene” approach). The dcanr package [103], implementing 11 different “global network” differential co-expression approaches, is particularly handy and well documented. An example of the kind of output obtained through this tool is shown in Figure 4, where the genes most significantly increasing their co-expression between normal breast and breast tumours in the TCGA (The Cancer Genome Atlas) cohort, identified via the z-score algorithm [61], are represented in a network with Cytoscape [101] (Figure 4A). Statistics on a network’s properties indicate overall differences in connectivity between the two conditions, showing that the differential network follows a power law, with a few genes having many connections and many genes having a few connections (Figure 4B). This network can be further dissected to gain module- and gene-level information through gene clustering and node-level statistics obtained, for example, with the igraph R package [104]. Modules can then be studied through functional enrichment with tools such as clusterProfiler [105] (Figure 4C). In this case, four modules with at least 20 nodes are detected, and the two largest modules show enrichment for immune-related and extracellular matrix/angiogenesis Gene Ontology categories, respectively. Alternatively, modules can be found via methods explicitly designed for differential modules’ detection, such as DiffCoex, which builds upon the widely used gene co-expression reconstruction method WGCNA (Weighted Gene Co-expression Network Analysis) [106]. DiffCoex, also implemented in the dcanr package, takes as input two expression matrices (e.g., normal and tumour) and identifies differentially co-expressed modules via the clustering of genes based on correlation differences across conditions. Finally, differential edges displaying the strongest change in correlation can be selected from the whole differential co-expression network in a “gene-specific” perspective (Figure 4D). In this example, LZTS1, having 54 edges, is the top differentially connected gene, followed by PODNL1 (Figure 4A), and it shows a strong change in relationship with its network neighbor LAMA4, switching from negative to positive correlation in normal tissue and breast tumour, respectively (Figure 4D).

The performance of different methods has been compared in a few studies. In particular, amongst “global network” approaches, z-score and entropy-based methods have been shown to be the most reliable in reconstructing differences between synthetic networks differing in a regulator’s expression levels [103]. Local “gene-specific” approaches, taking into account only the first neighbours of genes for scoring, have higher performance in detecting disease-related genes when compared with global “gene-specific” approaches [107]. Moreover, preserving information about all edges without setting hard thresholds proved to be an advantage [108]. However, the different evaluation methods used cannot be directly compared, suggesting that further efforts would be needed to reach a consensus about evaluation standards.

## 4. Differential Co-Expression Networks in Cancer

The methods described above have been applied to cancer biology to: (i) compare global topological features of cancer networks at different stages (“global network” approach), (ii) identify groups of co-regulated genes or pathways that are dysregulated in cancer (“module-based” approach), (iii) find specific genes changing their connectivity in cancer network (“gene-specific” approach). Moreover, despite the most frequent comparisons being tumours at different stages, many kind of conditions have been contrasted (e.g., ER+ vs. ER- breast tumours, p53 mutated vs. p53 wild type).

### 4.1. Global Topological Features of Cancer Networks Show Increasingly High Entropy

From the analysis of topological differences between cancer and normal tissue co-expression networks, some general principles have emerged. In particular, network entropy (disorder), measured with different metrics, has been shown to increase in cancer [109,110,111], paralleling an overall decreased connectivity [112]. Interestingly, entropy is even higher in tumours that metastasise, at least in the breast [113]. This observation may help in explaining the underlying principles of cancer adaptability and resistance to perturbations, such as treatments with drugs and hypoxia. In fact, the entropy is correlated with the system’s robustness [114]. In line with this idea, cancer cell lines resistant to three different tyrosine kinase inhibitors have been shown to display higher network entropy than their sensitive counterparts [115]. This could be interpreted as cancer displaying a higher number of interconnections and possible regulatory relationships for each gene, which makes the whole network resistant to single nodes’ and edges’ disruptions. Moreover, the relationship between entropy and robustness can inform about promising drug targets, represented by low-entropy genes [109,116]; interestingly, entropy usually decreases for up-regulated genes in a cancer’s networks [109]. A second important concept arising from these studies is the differential usage of nodes and edges in cancer: a cancer’s networks tend to be less hub dependent, displaying signalling shortcuts in comparison with normal tissues [117,118]. These features, observed in 13 different cancer types, are suggestive of facilitated crosstalk between biological processes that are usually not interconnected, again supporting a higher robustness of cancer networks.

Although both higher entropy and connectivity between pathways can be interpreted as a weakening of tight regulatory rules, improving tumour adaptability, they could also reflect higher cellular heterogeneity. This idea has been confirmed by Park et al. [119], who assessed network entropy related to cells’ heterogeneity and the number of subclones, making use of single-cell data, tumour purity estimates and clonal evolution in xenograft models. Additionally, signalling entropy has been shown to be an estimate for tumour stemness [120], and to be a prognostic measure across several epithelial cancers.

Interpreting network entropy as linked with tumour heterogeneity would help in explaining why it often decreases in cancer at advanced stages [115], confirmed by the observation that initial tumour heterogeneity is subsequently reduced by clonal selection and expansion in the process of metastasis [121].

### 4.2. Pathways Dysregulated in Cancer

The detection of co-expression modules has been widely applied to retrieve gene categories relevant to cancer, identifying modules shared across cancer types [29]. Nevertheless, condition-specific modules detected through differential co-expression allow the study of features characterising, for example, a disease state or different stages of the same disease, and have been shown to outperform single-condition co-expression modules in identifying characterising features of the studied biological system [122]. The analysis of the differential co-regulation of groups of genes (modules or pathways) has been carried out either exclusively using unbiased network properties or feeding prior knowledge-related gene lists to the network structure. This latter approach, despite its higher robustness and interpretability, has been employed only in a few studies [82,84,88,123,124] and needs to be better explored, while the unbiased analysis has also been amply applied to cancer, as described thoroughly below.

DC modules in cancer biology have been applied to the comparison of tumour and normal tissue, identifying tumour-specific modules in hepatocellular carcinoma, uveal melanoma and ovarian and prostate cancer [125,126,127]. In 12 cancer types, cancer-specific modules were shown to have prognostic value [128], while three independent studies reported immune response-related modules to be differentially co-regulated between ER+ and ER- breast cancer [42,103,129]. Immune-related modules are also differentially connected in non-small cell lung cancer when compared to normal tissue [70], possibly regulated through a miRNA-mediated mechanism. Interestingly, an independent report found an enrichment for targets of miRNAs related with cancer in two co-expression modules more strongly connected in lung cancer than in normal tissue [63]. Methods to identify modules that are differentially co-expressed across multiple networks find a particularly interesting application in the study of cancer stage-specific regulatory relationships [65,130,131]. In breast cancer, dynamically co-regulated modules improve the prediction of stage, and their hubs are enriched in signalling protein domains [130]. This observation confirms the idea that context-specific hubs are signalling or regulatory molecules that tune the activity of constitutive hubs, grouped in modules of genes with similar functions [132]. Only in a few cases has validation of *in silico* predictions been provided. Recently, however, the potential of differential co-expression in driving testable hypotheses has been demonstrated in an inclusive analysis of astrocytoma progression that integrated mRNA expression, ChIP-seq and copy number variation (CNV) data [133]. Indeed, the authors were able to identify a cell cycle-enriched module predicted to be affected by resveratrol and to experimentally validate their prediction.

Finally, specific comparisons allow for the investigation of regulatory differences between tumours classified according to various parameters: survival time [134], angiogenic features [71], type of treatment [135] and genomic stress [136]. A comprehensive study compared high and low genomic stress tumours in 15 cancer types, identifying 101 modules activated by genomic stress based on CNV, expression data, and PPI networks [136]. Within these, up-regulated hubs have been proposed as non-oncogene addiction genes for further functional studies. In the same vein, the differential co-expression module multivariate analysis method MultiDCox has been applied to breast cancer, revealing gene sets associated with mutant p53, ER status and grade [95].

### 4.3. Differentially Co-Regulated Genes

In cancer biology, “single-gene” approaches have been applied to prostate [56,91], gastric [137], liver [125,138], bladder [139,140], thyroid [141] and lung [142] tumours and glioblastoma [40], comparing the connectivity of genes between normal and cancerous tissue. This led to the identification of several gene lists not shared between different cancer types, as also confirmed by a systematic study performed on 12 cancer types [128]. Despite the gene-centred approach, in all studies, the genes with the strongest evidence for differential connectivity were analysed as a whole and often corresponded to previously known and druggable targets [125,128,137]. Despite the lack of a systematic study to compare the functional enrichment of DC genes across cancer types, recurrent Gene Ontology categories comprise cell cycle, apoptosis, and immune system-related genes [56,125,138,141,142,143]. Depending on the researcher’s interest, the search for DC genes can be restricted to selected gene lists [144,145]. For example, focussing on metabolic genes, a signature of genes suggestive of mitochondrial dysfunction was found to be differentially connected in seven cancer types [145]. Again, this approach can inform about differences in connectivity, allowing the grouping of samples based on any feature of interest. The comparison of patients responsive and non-responsive to a specific treatment is particularly promising to understand the rearrangement of gene networks leading to drug resistance. Through this means, well-known genes have been confirmed to confer platinum resistance (e.g., CCNE2, AKT1 and MYC), and the additional role of FGFR1 and TSC2 has been proposed [52].

The “gene-specific” approach finds a particularly interesting application in investigating the gene network neighbourhood of a pre-specified gene of interest, such as the tumour suppressors p53 or PTEN [42,91]. The opposing roles of the same gene in different cellular contexts, a widespread feature of cancer genes (e.g., Wnt5a, TGFbeta or p63 context-specific oncogenic and tumour-suppressive activity [146,147,148]) can be elucidated by this means. Surprisingly, only NOTCH1 has been investigated by differential co-expression, in lung adenocarcinoma (LUAD) and lung squamous cell carcinoma (LUSC) [149], predicting and then validating it as a pro-proliferative factor in LUAD, while acting as a tumour suppressor in LUSC.

### 4.4. Regulatory Mechanisms

Differential co-expression indicates changes in regulatory relationships between genes. This change can be mediated by a transcriptional regulatory mechanism that can be directly appreciated in transcriptional data (e.g., expression levels of a transcription factor or cofactor), as assessed in a set of works aiming at reconstructing GRNs [62,98,99,100]. Alternatively, changes can pertain to a different layer of molecular regulation (e.g., microRNA, DNA methylation, genomic mutations) and are hence not directly derived from transcriptional data. In any case, the integration of diverse data types regarding different regulatory layers can greatly improve the discovery and interpretation of altered gene regulatory networks, as reported in a comparative study showing an improved performance of almost all tested methods when integrating mRNA and miRNA data [107].

Indeed, the most explored of these layers is the regulatory activity of miRNAs on mRNAs, lncRNAs and mRNA–lncRNA crosstalk. In fact, it has been shown that RNA molecules sharing miRNA response elements (MREs) can communicate with each other by competing for common miRNAs (competing endogenous RNAs–ceRNAs-) [150]. The mRNA–miRNA–lncRNA connection implies a correlation of mRNA–lncRNA in the presence of a common regulatory miRNA, and no correlation in its absence.

The influence of miRNA expression on mRNA–lncRNA crosstalk has been studied in esophageal squamous cell carcinoma (ESCC), comparing two intrinsic subtypes and identifying the “loss” of miR-186-mediated PVT1–mRNA and miR-26b-mediated LINC00240–mRNA crosstalk [151] as driver events of ESCC development. The loss of connectivity in ESCC with respect to normal tissue led to the identification of two lncRNAs whose functional relevance has been experimentally validated [57]. The ceRNA concept has also been central in the analysis of tumour vs. normal co-expression networks in breast cancer, leading to the identification of the lncRNA AC145110.1 that significantly changes its interactions with 127 mRNAs involved in cell growth [152]. Importantly, merging mRNA and miRNA high-throughput data, two independent studies showed that known cancer miRNAs tend to dysregulate their connections with targets in the tumour network [153,154], regulating proliferative functions.

To facilitate these investigations, the method LncSubpathway has been specifically designed to address the change in crosstalk between mRNAs and lncRNAs across conditions [155], identifying lncRNA-regulated modules, while MACPath [97] searches for pairs of pathways regulated by a ceRNA mechanism, integrating the additional knowledge of transcripts’ miRNA responsive elements.

Other regulatory layers have been less explored: only a few studies integrated epigenetic changes with biological networks (e.g., [156], combining differential methylation and PPIs), but no direct inference of epigenetically regulated co-expression modules has been proposed. Of note, specific methods have been generated to assess co-expression dependence on sequence variants [92,93,94,157,158], and applied to identify genetic variations potentially responsible for gene network rewiring in disease [158]. In cancer, this led to imply the TF Myc-associated factor X (MAX) in modulating ATF4 and CLOCK interaction [93]. Interestingly, this result was supported by MAX binding to ATF4 and CLOCK promoters.

Nevertheless, the integration of multiple data types is intriguing, especially since it allows for moving from the retrieval of a differential co-expression to the identification of a modulator of gene interactions. The initial results are promising, and further work in this direction, especially taking into account epigenetic and post-translational modifications, would be profitable.

## 5. Conclusions and Perspectives

Disentangling molecular interactions underlying cancer properties may help in understanding and predicting cancer behaviour, selecting therapeutic targets and efficacious drugs to be employed for treatments. In this, gene co-expression networks and, even more, *differential* co-expression networks have shown a strong potential. A wealth of different methods has been developed (Table 2, Table 3, Table 4 and Table 5), allowing for pinpointing gene relationships specific to cancer or to precise cancer types/stages.

The investigation of changes in overall network structure revealed some of the principles explaining cancer robustness to escape otherwise deadly treatments, linked to a higher network entropy that can be explained by the concomitant usage of more biological pathways compared with normal tissue, together with intra-tumour heterogeneity. This kind of data can inform about which tumours are more likely to respond to treatments, but also about which nodes (low-entropy genes) are more likely to be susceptible within the network.

The alterations identified in the modular structure and compactness of cancer networks can pinpoint the biological functions or molecular pathways more strongly affected by cancer transformation. Gene co-expression networks, in fact, have a modular structure where genes belonging to the same module often share functions. This approach can rely on previous knowledge of pathways’ components and structure, as defined, for example, in KEGG pathways and through PPI networks, but also unbiasedly define modules based on expression levels only.

Through these means, groups of genes changing their association across cancer stages have been identified, and, despite a lack of a consensus on the most altered pathways due to differences in methods and specific comparisons, immune-related modules seem to be recurrent across studies. This implies that the composition and function of the tumour microenvironment is a crucial feature of cancers, often showing stronger remodelling than the tumour cells’ gene networks. Thus, employing gene co-expression networks in cancer cell- or stroma-specific datasets could be a more sensitive alternative to the use of bulk tumour gene expression. In the absence of wide enough datasets for network construction, deconvolution methods [159] represent a useful means to reconstruct cell type-specific gene expression from whole tumours, which can then be fed into differential gene co-expression tools.

Many efforts have been devoted to the identification of pairs of genes changing their relationships in cancer, in a “gene-specific” approach. This approach is particularly suited for the prioritisation of therapeutic targets, often defined as the genes that most strongly change their interaction pattern in cancer. Nevertheless, for ease of interpretation, often top-ranking genes are studied as an ensemble to run functional enrichment analysis, confirming the recurrence of immune-related categories for DC genes.

Many gene lists have been proposed in different studies, but drawing a general conclusion from such scattered studies is, at the moment, impossible. Therefore, to make the most of these methods, meta-analyses comparing results from published research or systematic analyses on independent datasets and with different methods should be performed. In particular, very little research has been devoted to the comparison of the methods’ performance, with a few exceptions. The most comprehensive comparison of network-based approaches showed the highest performance of z-score and entropy-based methods, while local and hybrid single-gene methods proved superior to global single-gene methods in identifying disease-related genes [107].

Beyond simple differential co-expression, some data integration methods have been proposed to determine altered regulatory mechanisms explaining the differential interactions. Among these, miRNA-based modulation of co-expression has been the most extensively studied, with very little effort devoted to DNA sequence variations, methylome or phosphoproteome. Given the strong impact on molecular mechanisms’ identification and interpretation that transcriptome-based modulator inference has allowed, efforts in this direction would be beneficial.

A serious concern arising from this literature review is the almost complete lack of experimental validations or biological studies following the *in silico* prediction of therapeutic targets and fundamental pathways involved in cancer. Proofs of the biological validity of differential co-expression methods are compelling, and multidisciplinary works employing such approaches integrated with experimental molecular studies would be a valuable combination to discover cancer molecular mechanisms.

A proxy for direct experimental validation may be represented by genome-wide interference experiments, followed by transcriptomics or functional assays. At least three screenings (the DepMap, Score and DRIVE projects) have profiled the relevance of each gene for the survival of hundreds of cancer cell lines [160,161,162], while the impact on the transcriptome is provided by the Connectivity Map, even if on a restricted panel of cell lines [163]. These results can be compared with those obtained through drug screenings [163,164,165] combined with the knowledge of drugs’ targets, as annotated in drug target databases such as DrugBank and ChEMBL [166,167]. Moreover, high-content screenings with PhagoKinetic Tracks Assays provide additional information on genes’ importance for cancer cells’ migration [168,169,170].

Despite these resources not allowing a direct validation of gene networks’ rewiring, they can be profitably used to assess the functional relevance of a differential hub or to show a causal link between the expression of connected genes in the network.

In conclusion, as the availability of genome-wide data on gene and protein expression, protein–protein interactions, transcription factors binding, non-coding RNAs, SNPs, mutational analysis and epigenetic modifications is rapidly growing, it is becoming increasingly essential to enhance the power and reliability of computing methods aimed at faithfully reconstructing the cell environment *in silico*. Compared to the widely employed co-expression networks, differential co-expression networks confer the key advantage of allowing a more precise reconstruction of disease-specific gene interactions, depleted of aspecific basic cell functions. Improving the precision of these reconstructions also via experimental or clinical validation will lead to predictive models of increasing precision, allowing for the formulation of stringent unbiased hypotheses related to the molecular causes of cancers, their resistance to drugs and the identification of the best alternative therapeutic targets.

## Figures and Tables

**Figure 1 ijms-21-09461-f001:**
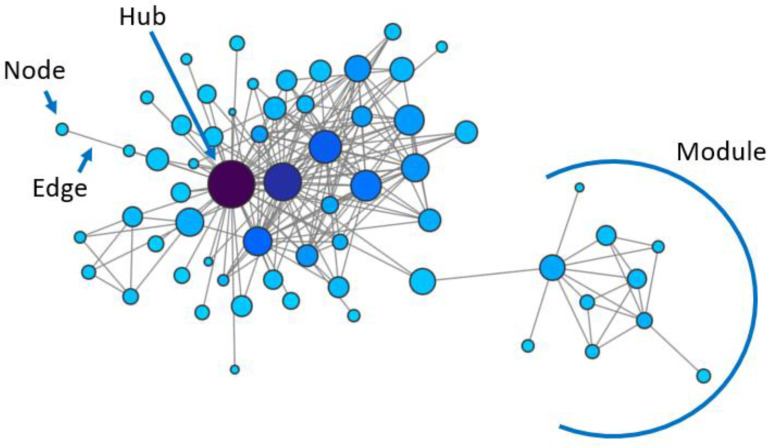
Example of a network, indicating nodes, edges, centrally located genes (hubs) and groups of tightly connected genes (modules).

**Figure 2 ijms-21-09461-f002:**
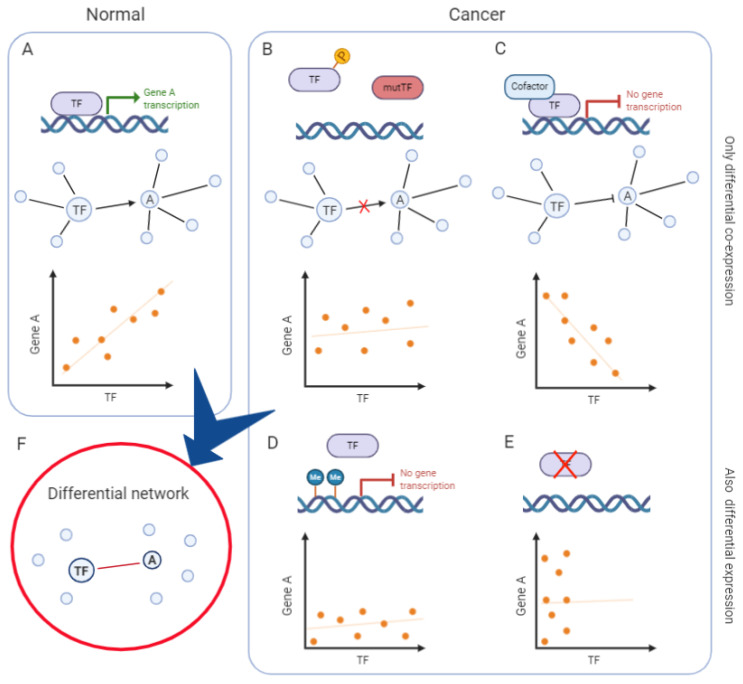
Mechanisms underlying differential co-expression and their impact on transcripts’ correlation. Example of a simplified network in normal tissue (**A**) and in cancer (**B**–**E**), whose comparison leads to defining a differential network (**F**). (**A**) Example of a simplified co-expression network in normal tissue, with a transcription factor (TF) activating its target, gene A. The TF and A are connected by an edge and their expression is positively correlated. (**B**) Post-translational modifications (e.g., phosphorylation) or mutations of the TF impede its binding to gene A promoter. Hence, the TF does not activate gene A expression: they are not transcriptionally correlated and therefore not linked in the co-expression network. (**C**) The presence of a cofactor alters the activity of the TF, that switches from acting as an activator to being a repressor of gene A expression. Hence, the TF and gene A are negatively correlated. In the network, they can be represented as having an inhibitory edge or as not being linked. (**D**) Epigenetic mechanisms (e.g., DNA methylation) inhibit the TF binding and gene A expression. The TF and gene A are not correlated and they are not linked in the co-expression network. (**E**) In the absence of TF expression, gene A expression does not depend on its regulatory activity. The TF and gene A are not correlated nor linked in the co-expression network. These regulatory mechanisms and others alter the relationships between the TF and gene A, changing their correlation at the transcriptional level. In some cases, one of the two genes is also differentially expressed (**D**,**E**), in other cases, only co-expression changes (**B**,**C**). Reconstructing the differential network between “normal” and “cancer” conditions, exemplified here, would lead to defining a network where only the TF and gene A are linked by an edge, indicating a decreased correlation in cancer (**F**).

**Figure 3 ijms-21-09461-f003:**
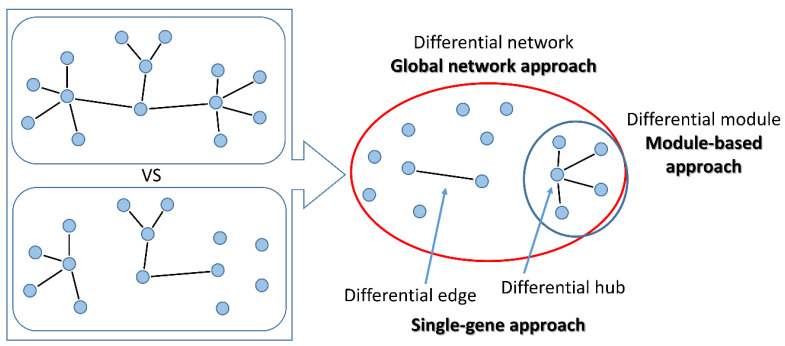
Methods for differential co-expression. Left: two conditions are compared. Right: with “global network” approaches, the overall differential network is built; “module-based” methods identify groups of differentially connected nodes; the “single-gene” approach searches for pairs of genes changing connections (differential edges) or single genes changing connectivity within the network (differential hubs).

**Figure 4 ijms-21-09461-f004:**
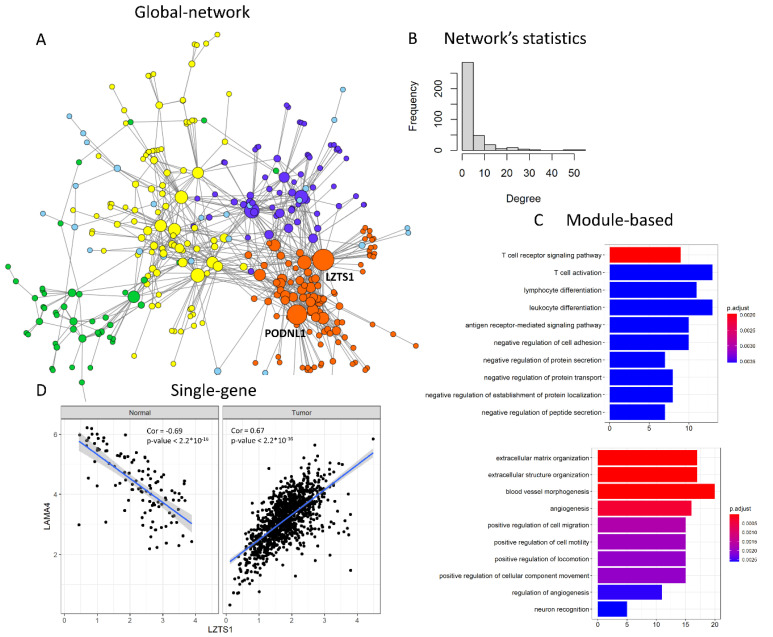
Example of differential co-expression analysis comparing TCGA normal tissue and breast tumour transcriptional data. (**A**) In a “global network” approach, the whole differentially co-expressed network is obtained. Here, only the top significantly differentially co-expressed genes are shown for simplicity (adjusted *p*-value < 10^−40^). Each node is a gene, colour indicates the cluster and size indicates the number of connections of that gene. The two nodes with the highest degree of differential connections are indicated (LZTS1 and PODNL1); (**B**) The differentially co-expressed network can be analysed as a whole, studying its topological properties, such as degree distribution; (**C**) In the global network, clusters (modules) can be identified and studied through functional enrichment, in a “module-based” approach. The top significantly enriched Gene Ontology (GO) categories for the two largest modules are shown in the two boxes, with bars indicating the number of genes belonging to each GO category and colour indicating the adjusted *p*-value; (**D**) Node and edge prioritisations allow for selecting specific genes or gene pairs with a particularly strong change in connectivity between the compared conditions: LZTS1 and LAMA4 are significantly negatively correlated in normal breast and switch to a positive correlation in breast tumours.

**Table 1 ijms-21-09461-t001:** Some of the most popular databases of biological networks.

Database	Type of Network	Link	Reference
DIP	PPI	https://dip.doe-mbi.ucla.edu/dip/Main.cgi	[11]
MINT	PPI	https://mint.bio.uniroma2.it/	[12]
IntAct	PPI	https://www.ebi.ac.uk/intact/	[13]
BioGRID	PPI	https://thebiogrid.org/	[14]
STRING	Various	https://string-db.org/	[15]
COXPRESdb	Co-expression	https://coxpresdb.jp/	[16]
SEEK	Co-expression	http://seek.princeton.edu/	[17]

**Table 2 ijms-21-09461-t002:** “Global network” methods that can also be employed for the “gene-specific” approach.

Method	Number of Conditions	Citation	Availability
DINGO	Multiple	[40]	CRAN R package (iDINGO)
Entropy	Two	[41]	Bioconductor R package (dcanr)
DGCA	Two	[42]	CRAN R package (DGCA)
Discordant	Two	[43]	Bioconductor R package (Discordant)
MAGIC	Two	[44]	Bioconductor R package (dcanr) MATLAB implementation at https://github.com/chiuyc/MAGIC
EBcoexpress	Two	[45]	Bioconductor R package (EBcoexpress)
GGM-based	Two	[46]	Bioconductor R package (dcanr)
LDGM	Two	[47]	Bioconductor R package (dcanr) Matlab implementation at https://github.com/ma-compbio/LDGM
Gill	Two	[48]	R package at http://www.somnathdatta.org/Supp/DNA
DDN	Two	[49]	MATLAB toolbox at http://www.cbil.ece.vt.edu/software.htm
Zhao	Two	[50]	
JDINAC	Two	[51]	R code at https://github.com/jijiadong/JDINAC
TDJGL	Two	[52]	R code at https://github.com/Zhangxf-ccnu/TDJGL
mlDNA	Two	[53]	CRAN R package (mlDNA, not maintained)
DCGL	Two	[54]	CRAN R package (DCGL)
DECODE	Two	[55]	CRAN R package (DECODE)
SIG method	Two	[56]	
Discordant	Two	[43]	Bioconductor R package (discordant)
DCN	Two	[57]	R package at https://github.com/weiliu123/DCN-package
TCDV	Two	[58]	
BFDCA	Two	[59]	R package at http://dx.doi.org/10.17632/jdz4vtvnm3.1
CODC	Two	[60]	R package at https://github.com/Snehalikalall/CODC
z-score	Two/De novo	[61]	Bioconductor R package (dcanr)
MINDy	De novo	[62]	Bioconductor R package (dcanr)

**Table 3 ijms-21-09461-t003:** “Module-based” methods.

**Method**	**Module Definition**	**# of Conditions**	**Citation**	**Availability**
DICER	Unbiased	Multiple	[63]	Bioconductor R package (dcanr) Java software at http://acgt.cs.tau.ac.il/dicer/
DiffCoEx	Unbiased	Multiple	[64]	R package at https://github.com/ddeweerd/MODifieRDev.git Bioconductor R package (dcanr)
M-Modules	Unbiased	Multiple	[65]	
NIPD	Unbiased	Multiple	[66]	
C3D	Unbiased	Multiple	[67]	
CoXpress	Unbiased	Two	[68]	R package at http://coxpress.sourceforge.net/
DiffCorr	Unbiased	Two	[69]	CRAN R package (DiffCorr)
ModMap	Unbiased	Two	[70]	Java executable at http://acgt.cs.tau.ac.il/modmap/
ALPACA	Unbiased	Two	[71]	R package at https://github.com/meghapadi/ALPACA
BFDCA	Unbiased	Two	[59]	R package at http://dx.doi.org/10.17632/jdz4vtvnm3.1
DiffCoMO	Unbiased	Two	[72]	
SCDA	Unbiased	Two	[73]	MATLAB implementation at http://vk.cs.umn.edu/SDC/
CODC	Unbiased	Two	[60]	R package at https://github.com/Snehalikalall/CODC
EgoNet	Unbiased	Two	[74]	
BicMix	Unbiased	Two	[75]	R package at https://github.com/chuangao/BicMix
MODA	Unbiased	Two	[76]	Bioconductor R package (MODA)
COSINE	Unbiased	Two	[77]	CRAN R package (COSINE)
DECluster	Unbiased	Two	[78]	
DEDC	Unbiased	Two	[79]	
DCN	Unbiased	Two	[57]	R package at https://github.com/weiliu123/DCN-package
Contrast Subgraph	Unbiased	Two	[80]	
DCIM	Unbiased	De novo	[81]	
GSCA	A priori	Two	[82]	Bioconductor R package (GSCA)
ScorePAGE	A priori	Two	[83]	
IB-GSA	A priori	Two	[84]	
Gill	A priori	Two	[48]	R package at http://www.somnathdatta.org/Supp/DNA
CoGa	A priori	Two	[85]	
dCoxS	A priori pairs of gene sets	Two	[86]	R function at http://www.snubi.org/publication/dCoxS/
MAGIC	A priori pairs of gene sets	Two	[44]	Bioconductor R package (dcanr) MATLAB implementation at https://github.com/chiuyc/MAGIC
ESEA	Structured pathway	Two	[87]	R package on CRAN (ESEA)
PWEA	Structured pathway	Two	[88]	R package on Bioconductor (ToPASeq)
KEDDY	Structured pathway	Two	[89]	Java implementation at https://sites.google.com/site/sjunggsm/keddy
kDDN	Structured pathway	Two	[90]	MATLAB implementation at http://www.cbil.ece.vt.edu/software.htm

**Table 4 ijms-21-09461-t004:** “Single-gene” methods.

Method	Description	# of Conditions	Citation	Availability
DEDC	Looks for the “best” DC gene	Two	[79]	
ECF	Given a pre-defined gene, selects others having differential co-expression with it	Two	[91]	CRAN R package (COSINE)
Gill	Given a pre-defined gene, tests whether its connectivity changes	Two	[48]	R package at http://www.somnathdatta.org/Supp/DNA

**Table 5 ijms-21-09461-t005:** Methods integrating co-expression and other sources of information.

Method	Description	Citation	Availability
FTGI	Integrates co-expression and SNPs	[94]	Bioconductor R package (dcanr)
MultiDCox	Multivariate. Identifies variables correlated with differential co-expression	[95]	R package at https://github.com/lianyh/MultiDCoX
dcVar	Differential co-expression based on sequence variants	[93]	Linux command-line tool at http://insilico.utulsa.edu/dcVar.php
wgLASSO	Integrates co-expression with PPIs	[96]	
MACPath	Integrates co-expression with annotation of miRNA-responsive elements	[97]	Python code at https://github.com/thejustpark/MACPath

**Table 6 ijms-21-09461-t006:** Methods to infer modulators of gene co-expression.

Method	Citation	Availability
z-score	[61]	Bioconductor R package (dcanr)
Mimosa	[98]	
GIMLET	[99]	R package at https://github.com/tshimam/GIMLET
MINDy	[62]	Bioconductor R package (dcanr) MINDy module in GenePattern
GEM	[100]	Implementation at https://sourceforge.net/projects/modulators

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
