# Peer review of "Differential Co-Expression Analyses Allow the Identification of Critical Signalling Pathways Altered during Tumour Transformation and Progression"

_ijms, 2020, doi:10.3390/ijms21249461_

Round 1

Reviewer 1 Report

In the review ‘Differential co-expression analyses allow to identify critical signaling pathways altered during tumor transformation and progression’, the authors review the literature on differential co-expression methods, divide them into broad types, and discuss applications in cancer biology.

The introduction to biological and gene co-expression networks, followed by the main proposed methods for study of differential co-expression are quite helpful for a user new to the field, and looking for an overview.

The pros and cons of each type of method are very insightful as well, also the errors in the referencing adds to a bit of confusion and lack of supporting data. (should be an easy fix, I’m assuming just a last minute mistake)

It is an interesting observation that immune-related modules are recurrent across studies, and this could be kept in mind during analysis. The knowledge confers a huge advantage over bulk gene expression experiments, that have become so commonplace today.

While overall a pretty comprehensive review, I would suggest, in each section, including one or two main tools in the main text and explaining using their example. It could be better for the new reader to know the mainstream tools and their usage, rather than be overwhelmed with a huge list of everything available.

In the abstract, authors mention ‘many conditions have been compared’ (line 22-23) : It would be beneficial to the reader to reword conditions to explain the context. Does it mean disease conditions?

It would be nice if the the authors can discuss about how their review is different or compares to (Sipko van Dam, Urmo Võsa, Adriaan van der Graaf, Lude Franke, João Pedro de Magalhães, Gene co-expression analysis for functional classification and gene–disease predictions, Briefings in Bioinformatics, Volume 19, Issue 4, July 2018, Pages 575–592, https://doi.org/10.1093/bib/bbw139) besides the specific cancer biology context? Given the similar landscape.

Overall, I believe this to be a well-researched and laid out manuscript, with a good presentation of the types and methods of co-expression analyses available, and outlining their potential in cancer research. I feel it could be made a little more crisp to drive home the main point in the Abstract, Section 4 and the Conclusion.

Author Response

Response to Reviewer 1 Comments

Point 1: The errors in the referencing adds to a bit of confusion and lack of supporting data.

Response 1: We apologize for the mistake, the referencing to figures and tables have now been fixed.

Point 2: While overall a pretty comprehensive review, I would suggest, in each section, including one or two main tools in the main text and explaining using their example. It could be better for the new reader to know the mainstream tools and their usage, rather than be overwhelmed with a huge list of everything available.

Response 2: To help the reader approach differential co-expression methods, we have added a paragraph detailing input and output of methods of the three categories (global network, module-based, single-gene). In the same paragraph and in a related figure, we describe the usage of a particularly well documented R package on breast cancer data (lines 230-256; Figure 4). With this simple example, we hope to provide the reader an overview of the different levels of analysis that can be performed on gene networks and on the insights they allow.

Point 3: In the abstract, authors mention ‘many conditions have been compared’ (line 22-23): It would be beneficial to the reader to reword conditions to explain the context. Does it mean disease conditions?

Response 3: We specified we are referring to normal vs disease conditions and different tumor stages (line 26).

Point 4: It would be nice if the the authors can discuss about how their review is different or compares to (Sipko van Dam, Urmo Võsa, Adriaan van der Graaf, Lude Franke, João Pedro de Magalhães, Gene co-expression analysis for functional classification and gene–disease predictions, Briefings in Bioinformatics, Volume 19, Issue 4, July 2018, Pages 575–592, https://doi.org/10.1093/bib/bbw139) besides the specific cancer biology context? Given the similar landscape.

Response 4: Indeed, the main difference between our and previous reviews is the cancer biology context. Nevertheless, from the methodological point of view we have extended Dam and co-authors’ discussion beyond module-based methods by describing the additional “global network” and “single-gene” levels of analysis. We have commented on these differences in lines 144-148.

Point 5: I feel it could be made a little more crisp to drive home the main point in the Abstract, Section 4 and the Conclusion.

Response 5: We did our best to improve abstract’s effectiveness (lines 14-35) and conclusions’ strength by commenting on the advantages of differential co-expression with respect to standard co-expression (lines 501-503). Moreover, we believe that the additional real-data example in Section 3 (lines 230-256) will make it more evident how the applications of differential co-expression to cancer biology, described in Section 4, can be biologically insightful.

Reviewer 2 Report

Major considerations-

Title- "Differential co-expression analyses allow to identify...." is awkward grammar.  Better to change the title to something like "Differential co-expression analyses permit identification of critical....."

The review is quite thorough in listing database resources, but what seems to be lacking is the next step.  What would the reader do when they access the databases listed.  How do they perform the correlative analyses and which analyses are useful to extrapolate and compare information.  It would be helpful to have a figure or table to show scatter plots of expression values, Z transformed correlations or correlations of genes in clusters to guide the reader in interpreting the data they have accessed.

Minor issue- Throughout the manuscript, "error source, not found" is labeled on lines 81, 167,168, 193,194, 196, 200, 219, 386

on line 169, Figure 3 is listed alone, not sure where it belongs. 

Author Response

Response to Reviewer 2 Comments

Point 1: Title- "Differential co-expression analyses allow to identify...." is awkward grammar.  Better to change the title to something like "Differential co-expression analyses permit identification of critical....".

Response 1: We changed the title to “Differential co-expression analyses allow the identification of critical signalling pathways altered during tumor transformation and progression”

Point 2: The review is quite thorough in listing database resources, but what seems to be lacking is the next step.  What would the reader do when they access the databases listed.  How do they perform the correlative analyses and which analyses are useful to extrapolate and compare information.  It would be helpful to have a figure or table to show scatter plots of expression values, Z transformed correlations or correlations of genes in clusters to guide the reader in interpreting the data they have accessed.

Response 2: To help the reader approach differential co-expression methods, we have added a paragraph and a figure describing the usage of a particularly well documented R package on breast cancer data (lines 230-256; Figure 4). We hope that this simple real-data example will complement the exemplificative scatter plots of Figure 2, providing an overview of the analysis workflow, and on the insights that differential co-expression networks allow.

Point 3: Minor issue- Throughout the manuscript, "error source, not found" is labeled on lines 81, 167,168, 193,194, 196, 200, 219, 386
on line 169, Figure 3 is listed alone, not sure where it belongs. 

Response 3: We apologize for the mistake, the references to tables and figures have now been fixed.
